# Losing ground in the field: An exploratory analysis of the relationship between work and mental health amongst women in conflict affected Democratic Republic of the Congo

**Julia Vaillant**[‡], **Naira Kalra**[‡*], **Alev Gurbuz Cuneo**[‡], **Léa Rouanet**[‡]

Gender Innovation Lab, Office of the Chief Economist, Africa Region, World Bank, Washington, D.C., United States of America

‡ These authors contributed equally to the manuscript. JV, NK, AGC and LR are joint first authors. Authorship is in random order.
* nkalra1@worldbank.org, nairakalra@gmail.com

**Data Availability Statement:** All relevant data are within the paper and its Supporting Information files.

## Abstract

### Background

Conflict affected populations, in particular women in such settings, face an increased risk of developing mental health disorders as well as well as economic vulnerability and reduced productivity. However, the link between the two has rarely been studied.

### Data and methods

The data in this paper come from a cross-sectional dataset (n = 1053) and a panel dataset of (n = 499) women suffering from post-traumatic stress disorder (PTSD) in eastern Democratic Republic of the Congo. This paper investigates the association between mental health disorders (PTSD, depression and/or anxiety) and employment for women in a conflict-affected setting.

### Results

The study finds that worsened local functioning is associated with reduced likelihood of working, earnings, and engagement in paid work. Reduction in probable depression and/or anxiety and PTSD are both associated with increased likelihood of engaging in paid work compared to unpaid work. Reduction in probable depression and/or anxiety is also associated with engaging in a secondary economic activity, as well as with higher productivity. However, when controlling for daily (local) functioning impairment, the primary pathway through which mental health may impact working, we detect a positive relationship between work or working hours and increased symptoms of PTSD and depression and/or anxiety. Working women with worse PTSD and depression and/or anxiety symptoms are also less likely to be self-employed, especially in an off-farm setting, and more likely to be engaged in farming.

**Funding:** The authors received no specific funding for this work. This research group is supported by the World Bank Group's Umbrella Facility for Gender Equality. The Umbrella Facility for Gender Equality (UFGE) is a multi-donor facility designed to strengthen awareness, knowledge, and capacity for gender-informed policy making. Funding is made possible through generous contributions from the governments of Australia, Canada, Denmark, Finland, Germany, Iceland, Norway, Spain, Sweden, Switzerland, the United Kingdom, and the United States. We are grateful to the World Bank Group's Nordic Trust Fund, and the Swiss Development Cooperation, for financial support.

**Competing interests:** The authors have declared that no competing interests exist

## Conclusion

A complex relationship between working and mental health emerges. Our findings also suggest that in this population farming, particularly farm-based wage work, is positively associated with worse mental health even after accounting for wealth and other relevant socio-demographic factors. These findings highlight the importance of paying close attention to the mental health of beneficiaries of livelihood support projects in post-conflict settings, where the relationship between mental health and employment is not straightforward.

## Introduction

Mental disorders are amongst the top three leading causes of disability worldwide and can directly impact an individual's capacity to work by deteriorating their physical, social, and cognitive functioning [1, 2]. In terms of prevalence, conflict-affected populations are at extremely high risk of experiencing mental disorders, they remain vulnerable to developing post-traumatic stress disorder (PTSD), depression, and anxiety at rates much higher than the general population [3]. On average, approximately one in four conflict affected individuals suffers from PTSD, depression and/or anxiety and these rates are typically higher in women than in men [3, 4].

In addition to their impact on well-being, mental health disorders are also costly, when they negatively affect a person's ability to generate income for themselves and their families. Several studies in the general population, in high income countries, have attempted to longitudinally explore the link between common mental disorders (depression and anxiety) and employment. More recently, research by Flint et al. [5] on a British sample, by Mitra and Jones [6] using data from the United States, by Jefferies et al. [7] on Europe and Chile, and by Olesen et al. [8] using an Australian sample explored this bi-directional relationship. These longitudinal studies find an association between common mental disorders and employment where improvements in mental well-being are associated with employment. The increase in income from employment can in turn improve mental well-being [8].

There is relatively less literature on the links between probable PTSD and economic well-being in the general population. These studies, mostly carried out in western contexts, tend to focus on Vietnam war veterans in the US. They find an association between severity of symptoms and reduced probability of working [9, 10], as well as an association between anxiety, major depressive disorders and employment and hourly wages [10].

Mental health is not only associated with reduced employment, but also with certain types of employment and quality of work. Smith et al. [9] found that among employed war veterans suffering from severe PTSD, those working in sales or clerical jobs had worse PTSD symptoms. A recent global systematic review of research on mental health and farming found that most studies indicate a stronger likelihood of mental disorders amongst farmers and farm owners compared to the general population [11]. Most of those studies were from developed countries. Butterworth et al. [12] find that jobs with greater uncertainty, lower skill, and greater adversity are no better in terms of mental health than being unemployed, despite their economic benefits. At the same time, drawing upon psychological theory and experiments that find a positive impact of physical activity on chronic pain and PTSD, research from post-conflict Mozambique postulates that physical labor, social connectedness, structure and routine during at least some cycles of agricultural work can reduce symptoms of PTSD [13].

Despite the increased vulnerability of conflict-affected populations, there are substantial gaps in the literature on the relationship between employment and both PTSD and depression and anxiety. Only a handful of studies have explored this relationship amongst women in

conflict-affected settings, and the findings are mixed. In 2010, a systematic review of factors influencing the mental well-being of conflict-affected populations [14] identified only one single study that explored the association between employment and mental health. This study uses a cross-sectional dataset from Bosnia and Herzegovina and found that compared to those who were working, those who were incapable of working had worse mental well-being [15]. Studies have mostly looked at the impact of assets [16] or savings groups [17] on symptoms of depression, anxiety, and PTSD, with some showing improvement in symptoms and others showing no impact. Similarly, the impact of business skills training and cash transfers on depression in conflict-affected women (and some men) in Uganda found no impact on depression [18]. However, the impact of an intervention that provides goods or skills is not necessarily comparable with the impact of being employed as the latter can have effects on mental health above and beyond gains from increased wealth. Further research on the association between employment, productivity, occupation type, and mental health in conflict-affected populations in lower middle-income countries is needed.

### Study context and aims

The DRC has been a conflict-affected country since the 1990's and continues to be one of the poorest countries in the world with roughly 63 percent of its population living below the poverty line [19].Women in eastern DRC are particularly vulnerable to violence and displacement due to persistent conflict and insecurity, with more than half of them experiencing physical or sexual violence at some point in their lives [20]. Gender-based violence survivors have described experiences of a range of mental health and psychosocial problems, such as anxiety, depression, and aggression [21]. A 2010 prevalence study carried out in North and South Kivu provinces and Ituri district in the DRC found that half of the adult respondents exhibited symptoms indicative of PTSD [22]. In addition to the mental health consequences of GBV, female survivors of violence and conflict are also at risk of exclusion from economic opportunity and thus experience an increased risk of drifting into or remaining in poverty [23, 24].

A 2018 report analyzing job creation and its main challenges in the DRC economy finds that self-employment and unpaid work in the agriculture sector in rural areas are the dominant forms of employment [25]. According to this report, while women's labor participation in the DRC is high (70.7%), they are more likely to have a transient employment status and work in the informal sector. It is not clear from existing literature if there is a relationship between mental health and these various aspects of employment among women in conflict-affected settings, especially in DRC.

This paper aims to investigate the association between mental health outcomes such as PTSD, depression, and anxiety, and labor market outcomes such as employment status and productivity in eastern DRC. It uses data collected for a randomized controlled trial of a mental health intervention provided to women suffering from PTSD. The study will leverage the panel structure of the data to unpack the relationship between changes in mental health and employment, earnings, and productivity. Additionally, the study will explore the association between mental health and the sector and type of employment in which women engage.

In 2012, approximately 61% of the working population was self-employed and 20.5% were unpaid workers; where 16.6% had wage

## Materials and methods

### Data

Data were collected as part of a randomized controlled trial of Narrative Exposure Therapy (NET) in eastern DRC [26]. Baseline data were collected between June 2017 and October 2019,

and follow-up data collection was conducted between January 2018 and May 2020. All data were collected electronically. Participants were enrolled in the study as they sought care in hospitals, health centers, and community-based organizations and were blind to treatment assignment at baseline. Eligibility criteria included: being a woman aged 18 or above, screened by a professional, and diagnosed with PTSD disorder. The PTSD Symptom Scale–Interview for DSM-5 (PSS-I-5) was used to make a diagnosis of PTSD for this study [27]. Although being a survivor of conflict-related violence was not an eligibility condition, the intervention was designed to address the mental health needs of women survivors of sexual violence. Written informed consent was collected at study registration, baseline and follow-up data collection.

The study protocol received approval from Health Media Lab with approval number 404WBG17, and ethical approval was obtained from the National Ethic Committee for Health in the DRC (Comité National d'Éthique de la Santé) with approval number 035/CNES/BN/PMMF/2016 on October 29[th], 2016.

In total 1,053 women completed the baseline survey in the city of Goma in North Kivu (n = 89), in the health zones of Rutshuru and Kirotshe in North Kivu (n = 534), and in the health zones of Kaniola and Minova in South Kivu (n = 430). A health zone is a decentralized administrative unit of the DRC which operates in accordance with the guidelines and standards set by the central level of the health system. Except for the city of Goma, the health zones in our sample are predominantly rural and poor. In each health zone, participants were identified when they sought care either in Community Based Organizations (CBOs), about 78% of the sample, in health centers, or at the Health Africa Center of Excellence in the city of Goma. Health centers are facilities at the lowest level of the heath system in the DRC. They also offer psychosocial assistance and legal support to survivors. NET counselors in health centers are mostly nurses and midwives. CBOs are village women's associations that offer assistance and reintegration services to survivors and socioeconomic services such as Village Savings and Loans Associations and livelihoods groups.

A follow-up survey was conducted approximately 6 months after baseline data collection. In the analysis, we consider the subsample of women who were randomized into not receiving NET treatment after the baseline survey (hereafter called the "no-therapy group"). Using the subsamples of these non-treated individuals both at baseline (n = 528) and at follow-up (n = 499), we create a panel dataset that is used in individual fixed effect models in the analysis. The attrition rate between baseline and follow-up was 6.1%. Main reasons for attrition were not being found (10 cases) and having moved out of the area (13 cases). One refusal case was recorded at follow-up. The entire baseline sample, combining both the treatment and the "no-therapy group", is also used for cross-sectional analyses on the link between type of work and mental health of the participants.

The "no-therapy group" received the standard-of-care provided to all women who sought care at Heal Africa, in health centers, or in CBOs. The standard-of-care includes case management services, such individual and collective counselling, referral to economic and legal support services, and health services in the case of health centers and Heal Africa.

## Main indicators

**Mental health outcomes.**   The key indicators for mental health measure (i) PTSD severity and (ii) combined depression and anxiety severity. Following Bass et al. [28] and Bass et al. [17], we assess the severity of PTSD using the Harvard Trauma Questionnaire (HTQ-16) [29] and depression and anxiety symptoms using locally adapted versions of the Hopkins Symptom Checklist-25 (HSCL-25) [30]. These checklists had been validated in studies from the DRC using similar populations of adult women sexual and gender-based violence survivors [28].

Both checklists ask the respondents how frequently they experienced a list of related symptoms over the last 4 weeks using a 4-point Likert scale (0 = not at all, 1 = a little bit, 2 = a moderate amount, 3 = a lot). Each score is constructed as the average of per-item scores, ranging from 0 to 3 where higher scores indicate worse symptoms. In addition to the continuous indicators, we also consider constructs that indicate probable PTSD and probable depression and/or anxiety, which are equal to unity if the relevant severity score is equal to or higher than the cut-off point, set at 1.75 for both indicators. Based on data from other conflict-affected populations, an average PTSD Checklist score of 1.75 or higher is predictive of clinically significant PTSD (Bass, et al., 2013). An average HSCL-25 score equal to or higher than 1.75 is predictive of clinically significant depression or anxiety [28, 31].

We also construct measures of three PTSD symptom clusters [32]. We use symptom clusters for exploratory purposes rather than diagnostic purposes. Given that in conflict affected populations both DSM IV and DSM V conceptualizations of PTSD seem to fit the data equally well [33], we explore the symptom clusters of re-experiencing, avoidance, and hyperarousal symptoms from the DSM IV criteria. While Michalopoulos et al. [32] find that a four factor 'numbing' model may fit the data slightly better in the DRC, the three-factor model in their study also had a good fit. As we did not wish to differentiate between active avoidance and numbing symptom clusters, we chose to explore avoidance overall. This allows us to situate our study within the existing literature that largely looks at these three symptom clusters. The continuous variables for these symptom clusters are used for exploratory purposes and the econometric regressions follow the same specifications as the primary mental health indicators.

In line with Bass et al. [28], we measure survivors' functional impairment using a local functioning impairment index. The respondents are asked to what extent they are experiencing difficulties in performing important daily activities; such as farming, income generating activities, housework, socializing, focusing on their tasks and responsibilities in the last 4 weeks. The local functioning impairment (LFI) index is created by taking the average of 14 item scores rated on a 5-point Likert scale with values ranging from "0 = no difficulty" to "4 = so difficult that she couldn't do it", where higher scores indicate greater impairment in local functioning. For ease of comparability, we used the standardized versions of the mental health indices and LFI index in our analysis.

## Employment outcomes

We consider several measures for the respondents' employment. For labor force participation, we construct a binary indicator that is equal to 1 if the respondent reported any kind of work in the last 7 days. Work includes all non-domestic work both paid and unpaid, self-employed or hired work, and work for the family business or farm. We consider two versions of the paid work indicator. The first indicator for paid work is a binary variable that equals 1 if the respondent worked and generated positive earnings in the last 7 days and equals 0 if she worked but didn't report earnings. The second indicator for paid work is a binary variable that equals 1 if the respondent worked and generated positive earnings in the last 7 days and equals 0 if she didn't work at all. Similarly, the indicator for unpaid work is a binary variable that equals 1 if the respondent worked but did not generate earnings in the last 7 days and equals 0 if she didn't work. We also look at whether the respondent is performing a secondary activity in addition to their main activity.

Next, we consider survivors' productivity and earnings. Among the respondents who had worked in the last 7 days, we create an earnings indicator that is the total amount of the respondent's earnings from her main and secondary activities in the last 7 days in United

States Dollar (USD). We also consider total hours of work she spent in her main and secondary activities during the last 7 days. Using these two measures, we construct a productivity measure that assesses earnings per hour by dividing total earnings by total hours worked in the last 7 days. Earnings and earnings per hour are winsorized at the top 1% level to exclude outliers. All results are robust to other transformations of these measures and available upon request.

Lastly, we construct indicators on the status and sector of respondents' main activity among women who reported they were engaged with paid work in the last 6 months (n = 885). The employment status can be (i) self-employed, (ii) wage-worker, or (iii) working in the farming sector as self-employed or as a wage worker. Self-employment (respectively wage work) and farming are not mutually exclusive, but there is no overlap between self-employment and wage work. We also create four mutually exclusive dummy indicators that are equal to one if the respondent works (iv) in the farming sector as self-employed (*on-farm and self-employed)*, (v) in a non-farm sector as self-employed (*off-farm and self-employed*), (vi) in the farming sector as wage-worker (*on-farm and wage-worker*), and (vii) in a non-farm sector as wage-worker (*off-farm and wage-worker*). All the indicators are equal to 0 if the respondent is working in a different sector/status, conditional on having paid worked in the last 6 months.

### Econometric specification

To estimate the association between survivor's mental health and labor market outcomes, we first run individual fixed effect panel regressions using the two available data points for women in the no-therapy group by estimating the following equation:

$$y_{it} = \alpha_i + \alpha_1 \, MH_{it} + t_i + \upsilon_{it} \tag{1}$$

where $y_{it}$ is the labor market outcome of interest measured for participant $i$ at time $t$, and $MH_{it}$ is a mental health indicator for participant $i$ at time $t$, where $t = \{1,2\}$. $\alpha_i$ is the individual specific intercept that captures heterogeneity across individuals and $t_i$ is the indicator variable for the time effect and is equal to 1 for a baseline data point, 0 for a follow-up data point.

We estimate this individual fixed effect model for the subset of labor market outcomes described above, specifically labor force participation and earnings. The panel regressions allow us to exploit change within individuals over time and thus eliminate unobserved time-invariant heterogeneity.

We are also interested in the sector and employment status as outcomes, but as there is little change in these variables between baseline and follow-up, a panel regression would not be meaningful here (see discussion of Table 6 in Section 4.1). A majority of women stay in the same sector over time, making the sample size on which to identify correlations over time too small and thus a cross-section analysis better suited for this question. For these outcomes, we estimate the following OLS model using the baseline data only, for all women (in the treatment and no-therapy groups):

$$y_i = \beta_0 + \beta_1 \, MH_i + \beta_2 \, X_i + \beta_3 \, \pi_i + \varepsilon_i \tag{2}$$

where $y_i$ is the labor market outcome of interest measured at baseline for participant $i$, and $MH_i$ is a mental health indicator. $X_i$ is a vector of control variables that include respondent's age, education, marital status, whether she is the household head, and household size. Since employment status and sector could also be correlated with poverty and wealth, which in turn may impact mental health measures directly, we control for asset and livestock ownership in regressions with employment status as the outcome variable. Similarly, we control for employment and sector indicators in the productivity and earnings regressions, as employment status and sector of work may impact mental health measures through their impact on earnings and

productivity. Lastly, $\pi_i$ is a vector of indicator variables for the respondent's health zone and study cohort which controls for location and time specific characteristics.

In these cross-sectional specifications, similar to the individual fixed effect model introduced above, we are not able to identify the direction of causality. In both specifications, the association estimated could result from two directions of causality. First, a worsening of the mental health of women can impair their functioning, and lead to a lower engagement in the labor force. Second, engaging in the labor force could directly impact the mental health of individuals, either positively or negatively, depending on their personal situation and working conditions. For this reason, in addition to our core specifications, we also estimate the same associations by adding respondent's local functioning impairment index as a control in Eqs (1) and (2). Functioning impairment is a key diagnostic criterion to determine the presence of a mental disorder [34]. Physical, social, and cognitive functioning are considered as the primary pathway through which mental illness impacts employment and other labor market outcomes [1, 2]. Therefore, controlling for this allows us to estimate the association between labor market and mental health measures net of the impact a respondents' level of PTSD or depression and/or anxiety might have on their functioning. By accounting for this key pathway between mental illness and labor market outcomes, we hope to shed light on how changes in employment and other labor market outcomes affect mental health indicators beyond their impact on functioning. While functioning is not a key pathway through which labor market outcomes impact mental health; employment, productivity and hours worked can still impact physical and social functioning negatively through reducing time availability for social activities or through work induced injury or pain. Therefore, our estimate of the association between employment and other labor market outcomes with mental health is also above and beyond the impact these may have on functioning impairment.

## Results

### Socio-epidemiological data

Study participants were on average 35.5 years old and had a low level of education with an average of 3.2 years of formal schooling (Table 1). Approximately 45 percent of the sample had never been to school. 44 percent of study participants are married or living with a partner, which is significantly lower than the national average of 64% [35]. Among those, the average age when they married or moved in with their current partner is 20. Around 41 percent of the women in the sample are the heads of their household and they have slightly fewer than four children under age 15 on average.

Approximately 70% of respondents in the sample meet the criteria for probable PTSD, and this number increases to 79.3% for probable depression or anxiety (Table 2). There is no significant difference between PTSD or HSCL-25 scores across groups of women with different working status in the last 7 days. However, we see a significantly greater local functioning impairment among women not working. There is also a significant difference in the re-experiencing PTSD symptom cluster, with women suffering from these PTSD symptoms being more likely to work. We see a general improvement of mental health over time, such that mental health disorders are less prevalent at follow-up than at baseline for the women in the no-therapy group (Table 3).

At baseline around 87% of the sample reported having worked, and 68% did paid work in the last 7 days (Table 4). This is higher than the average employment in both North Kivu (63.3%) and South Kivu (77.4%) regions of the DRC [35]. Nearly half of the respondents have a secondary activity in addition to their main activity. We see an increase in labor force participation over time: women are more likely to have worked in the last 7 days and to have a secondary activity at follow-up (Table 5).

**Table 1. Respondent characteristics at baseline.**

| | N | Mean (SD) / % |
|---|---|---|
| Age of the respondent (years) | 1053 | 35.49 (12.91) |
| Married or living with a partner | 1053 | 43.80% |
| Age when married/moved in with partner (years) | 457 | 19.99 (5.52) |
| Respondent has been in school | 1053 | 55.30% |
| Years of education of the respondent (years) | 1053 | 3.19 (3.83) |
| Respondent is the household head | 1053 | 40.80% |
| Head of household is a woman | 1053 | 45.20% |
| Total household size | 1053 | 6.66 (2.56) |
| Number of children (under age 15) | 1053 | 3.78 (2.11) |
| Respondent asset ownership index | 1053 | 2.12 (3.07) |
| Household asset ownership index | 1053 | 6.71 (6.72) |
| Total amount of savings in last 12 months in USD | 1053 | 5.31 (12.10) |
| Respondent is very likely to raise USD 36 in a month in case of emergency | 1051 | 8.3% |
| A household member had to skip a meal due to not enough food in last 7 days | 1052 | 77.3% |

Notes: Total amount of savings in last 12 months in USD variable is winsorized at the top 2.5% level.

Approximately half of the sample at baseline is self-employed, and 38% is a wage worker. Over time for the no-therapy group, more women work as self-employed, while less women work as wage workers, this decrease being driven by women in the farming sector (Table 5). About three quarters of the sample declare farming as their main activity, and this does not change significantly over time.

Women in our sample worked 32 hours a week at baseline and earned approximately USD 3.5 on average in the last week (Table 4). The no-therapy group women increased their earnings by USD 0.83 a week on average, without increasing the number of hours worked, which led to an increase in their earnings per hour (Table 5).

Next, we examine sector switches over time of women in the no-therapy group. A majority of the sample did not change sectors between baseline and follow-up: approximately 87% of

**Table 2. Mental health outcomes by work status at baseline, full sample.**

| | Not Worked | | Worked | | Total | | |
|---|---|---|---|---|---|---|---|
| | N | Mean (SD)/ % | N | Mean (SD)/ % | N | Mean (SD)/ % | Difference |
| | | (1) | | (2) | | (3) | (1)-(2) |
| PTSD checklist score | 133 | 1.86 (0.65) | 920 | 1.97 (0.57) | 1053 | 1.96 (0.58) | -0.118 |
| Probable PTSD | 133 | 63.20% | 920 | 71.20% | 1053 | 70.20% | -0.08 |
| HSCL-25 score | 133 | 2.10 (0.6) | 920 | 2.14 (0.54) | 1053 | 2.14 (0.54) | -0.042 |
| Probable depression or anxiety | 133 | 76.70% | 920 | 79.70% | 1053 | 79.30% | -0.03 |
| Local functioning impairment | 133 | 1.87 (0.8) | 920 | 1.34 (0.64) | 1053 | 1.41 (0.69) | 0.528*** |
| PTSD score—re-experiencing | 133 | 1.92 (0.84) | 920 | 2.14 (0.71) | 1053 | 2.11 (0.73) | -0.213** |
| PTSD score—avoidance | 133 | 1.81 (0.65) | 920 | 1.87 (0.61) | 1053 | 1.86 (0.61) | -0.064 |
| PTSD score—hyperarousal | 133 | 1.87 (0.75) | 920 | 1.99 (0.68) | 1053 | 1.97 (0.69) | -0.118 |

Notes: The value in the last column displays the differences in the means across the groups for the t-tests.

*Indicates significance at 10% level

** at 5% level

*** at 1% level.

**Table 3. Mental health outcomes for the no-therapy group at baseline and follow-up.**

| | Baseline | | Follow-up | | |
|---|---|---|---|---|---|
| | N | Mean (SD)/% | N | Mean (SD)/% | Difference |
| | | (1) | | (2) | (1)—(2) |
| PTSD checklist score | 528 | 1.95 (0.56) | 499 | 1.70 (0.64) | 0.251*** |
| Probable PTSD | 528 | 70.10% | 499 | 55.90% | 0.142*** |
| HSCL-25 score | 528 | 2.14 (0.54) | 499 | 1.82 (0.61) | 0.323*** |
| Probable depression or anxiety | 528 | 79.40% | 499 | 61.90% | 0.174*** |
| Local functioning impairment | 528 | 1.44 (0.70) | 499 | 1.20 (0.67) | 0.232*** |
| PTSD score—re-experiencing | 528 | 2.11 (0.71) | 499 | 1.77 (0.81) | 0.341*** |
| PTSD score—avoidance | 528 | 1.86 (0.60) | 499 | 1.66 (0.65) | 0.201*** |
| PTSD score—arousal | 528 | 1.96 (0.67) | 499 | 1.71 (0.76) | 0.247*** |

Notes: The value in the last column displays the differences in the means across the groups for the t-tests.

*Indicates significance at 10% level

** at 5% level

*** at 1% level.

the women engaged in farming activities at baseline stayed in the farming sector at follow-up (Table 6, Panel A). Similarly, 79% of women working as self-employed at baseline were still self-employed at follow-up, and around 52% of wage workers at baseline also worked as a wage worker at follow-up. Thus a cross-sectional approach using only baseline data is appropriate for the analysis of the relationship between sector of employment and mental health outcomes since sector switches are infrequent.

Lastly, we look at the balance between the women who received the treatment and the ones who did not in order to check whether the two different samples used in the panel and cross-

**Table 4. Labor market outcomes of interest at baseline, full sample.**

| | N | Mean (SD)/% |
|---|---|---|
| Worked in the last 7 days | 1053 | 87.4% |
| Worked in the last 6 months | 1053 | 94.9% |
| Paid work in the last 7 days (= 0 if not worked or unpaid work) | 1053 | 68.3% |
| Paid work in the last 7 days (= 0 if unpaid work) | 920 | 78.2% |
| Paid work in the last 7 days (= 0 if not worked)[1] | 852 | 84.4% |
| Unpaid work in the last 7 days (= 0 if not worked)[2] | 331 | 59.8% |
| Respondent has a secondary activity | 980 | 47.2% |
| Total earnings per hour in the last 7 days in USD | 876 | 0.13 (0.16) |
| Total earnings generated in the last 7 days in USD | 978 | 3.54 (4.93) |
| Total hours worked in the last 7 days | 920 | 32.23 (17.19) |
| Self-employed in the main activity[1] | 885 | 56.3% |
| Farming is the main activity[1] | 885 | 73.6% |
| On-farm and self-employed in the main activity[1] | 885 | 40.3% |
| Off-farm and self-employed in the main activity[1] | 885 | 15.91% |
| On-farm and wage-worker status in the main activity[1] | 885 | 33.2% |
| Off-farm and wage-worker status in the main activity[1] | 885 | 10.5% |

Notes: [1] Excludes unpaid work.

[2] Excludes paid work.

**Table 5. Labor market outcomes of interest for the no-therapy subsample at baseline and follow-up.**

| | N | Baseline Mean (SD)/% | N | Follow-up Mean (SD)/% | Difference |
|---|---|---|---|---|---|
| | | (1) | | (2) | (1)-(2) |
| Worked in the last 7 days | 528 | 88.1% | 457 | 91.90% | -0.038** |
| Worked in the last 6 months | 528 | 95.5% | 457 | 98.2% | -0.028** |
| Paid work in the last 7 days | 528 | 71.0% | 457 | 79.6% | 0.086*** |
| Paid work in the last 7 days (= 0 if unpaid work) | 465 | 80.1% | 462 | 78.8% | 0.019 |
| Paid work in the last 7 days (= 0 if not worked)[1] | 438 | 85.6% | 401 | 90.8% | -0.052** |
| Unpaid work in the last 7 days (= 0 if not worked)[2] | 152 | 58.6% | 92 | 59.8% | -0.012 |
| Respondent has a secondary activity | 493 | 47.9% | 443 | 62.80% | -0.149*** |
| Total earnings per hour in the last 7 days in USD | 447 | 0.13 (0.17) | 402 | 0.16 (0.21) | -0.029** |
| Total earnings generated in the last 7 days in USD | 493 | 3.45 (4.40) | 442 | 4.28 (5.88) | -0.831** |
| Total hours worked in the last 7 days | 465 | 32.11 (16.68) | 420 | 32.14 (17.36) | -0.031 |
| Self-employed in the main activity[1] | 454 | 56.4% | 404 | 63.9% | -0.075** |
| Farming is the main activity[1] | 454 | 74.4% | 404 | 73.0% | 0.014 |
| On-farm and self-employed in the main activity[1] | 454 | 41.6% | 404 | 47.0% | -0.054 |
| Off-farm and self-employed in the main activity[1] | 454 | 14.8% | 404 | 16.8% | -0.021 |
| On-farm and wage-worker status in the main activity[1] | 454 | 32.8% | 404 | 26.0% | 0.068** |
| Off-farm and wage-worker status in the main activity[1] | 454 | 10.8% | 404 | 10.1% | 0.006 |

Notes: The value in the last column displays the differences in the means across the groups for the t-tests.

*Indicates significance at 10% level

** at 5% level

*** at 1% level.

[1] Excludes unpaid work.

[2] Excludes paid work.

**Table 6. Sector and work status switches of the no-therapy sample over time.**

Panel A. Sector switch of the no-therapy sample over time

| | Baseline | Follow-up | Stayed in the same sector/status between baseline and follow-up | |
|---|---|---|---|---|
| | (N) | (N) | (N) | (%) |
| Farming | 329 | 326 | 285 | 86.6 |
| Self-employed | 223 | 253 | 177 | 79.4 |
| Wage worker | 166 | 142 | 86 | 51.8 |
| On-farm and self-employed | 162 | 186 | 119 | 73.5 |
| On-farm and wage worker | 124 | 102 | 54 | 43.5 |
| Off-farm and self-employed | 61 | 67 | 31 | 50.8 |
| Off-farm and wage worker | 42 | 40 | 14 | 33.3 |

Panel B. Work status switch of the no-therapy sample over time

| | Follow-up | | | |
|---|---|---|---|---|
| Baseline | Paid last 7d (= 0 if not worked) | Unpaid last 7d (= 0 if not worked) | Not worked last 7d | Total |
| Paid last 7d (= 0 if not worked) | 278 | 23 | 19 | 320 |
| Unpaid last 7d (= 0 if not worked) | 55 | 21 | 6 | 82 |
| Not worked last 7d | 31 | 11 | 11 | 53 |
| Total | 364 | 55 | 36 | 455 |

sectional regressions are comparable. The two groups are balanced at baseline among all mental health and employment outcomes, except paid work vs. unpaid work in the last 7 days. We find that more women in the no-therapy group engage in paid work in the last 7 days compared to the women in the treatment arm and this difference (5 percent points) is significant only at 10% (See S1 Table).

## Women's work and mental health

In Table 7, each cell shows the estimation result for the regression of one labor outcome on one mental health explanatory variable, as described in Eq (1). There is no significant association between doing any work and mental health outcomes, and there is no association between mental health outcomes and engaging in paid work relative to not working. However, among women who do work, those who switch from having probable PTSD at baseline to not having probable PTSD at follow-up are 7.7 percentage points more likely to switch from engaging in unpaid work at baseline to paid work at follow-up, that is a 9.5% increase compared to the baseline mean (column 2). The same holds for women whose HSCL-25 score drops below the 1.75 cutoff at follow-up: they are 8.1% points more likely to engage in paid rather than unpaid

**Table 7. Panel regression of women's work on mental health outcomes.**

| | (1) | (2) | (3) | (4) | (5) | (6) | (7) | (8) |
|---|---|---|---|---|---|---|---|---|
| | Worked in the last 7 days | Paid work last 7d (= 1 if worked with earnings; = 0 if worked w/o earnings) | Paid work last 7d (= 1 if worked with earnings; = 0 if not worked) | Unpaid work last 7d (= 1 if worked with no earnings; = 0 if not worked) | Respondent has a secondary activity | Total earnings per hour last 7d | Total earnings generated last 7d | Total hours worked last 7d |
| PTSD checklist score | 0.00260 | -0.0332 | 0.00421 | -0.000158 | -0.0341 | -0.00717 | 0.218 | 1.143 |
| | (0.0152) | (0.0222) | (0.0176) | (0.0734) | (0.0261) | (0.00956) | (0.251) | (0.829) |
| Probable PTSD | -0.0250 | -0.0771* | -0.0321 | -0.00471 | -0.0779 | -0.0127 | 0.549 | 2.110 |
| | (0.0293) | (0.0426) | (0.0341) | (0.142) | (0.0506) | (0.0182) | (0.487) | (1.593) |
| HSCL-25 score | -0.00575 | -0.0501** | 0.00287 | -0.0802 | -0.0417 | -0.00804 | 0.0475 | 0.229 |
| | (0.0171) | (0.0246) | (0.0197) | (0.0793) | (0.0295) | (0.0108) | (0.284) | (0.937) |
| Probable dep or anx | -0.0137 | -0.0812* | -0.0177 | -0.0254 | -0.153*** | -0.0350* | -0.230 | 1.800 |
| | (0.0323) | (0.0473) | (0.0376) | (0.152) | (0.0562) | (0.0200) | (0.545) | (1.775) |
| LFI index | -0.105*** | -0.0264 | -0.110*** | -0.175*** | -0.0559** | -0.00537 | -0.513** | -1.311 |
| | (0.0139) | (0.0226) | (0.0160) | (0.0569) | (0.0259) | (0.00979) | (0.250) | (0.872) |
| Observations | 985 | 927 | 839 | 244 | 936 | 849 | 935 | 885 |
| Individual Fixed Effects | Y | Y | Y | Y | Y | Y | Y | Y |
| Survey round dummy | Y | Y | Y | Y | Y | Y | Y | Y |
| Baseline Mean | 0.881 | 0.806 | 0.856 | 0.586 | 0.479 | 0.130 | 3.446 | 32.11 |

Notes: Each cell shows the estimation result for the regression of one labor outcome on one mental health explanatory variable. The regressions reported in the same column have the same sample size given in the observations row. Panel regressions are using baseline and follow-up observations for the no-therapy group, and controls for individual fixed effects ("Y" stands for yes for controlling for individual fixed effects and survey round dummy). Mental health outcomes and LFI index are standardized. Earning values are winsorized at the top 1% level. Standard errors in parentheses.

* $p < 0.1$

** $p < 0.05$

*** $p < 0.01$.

work. A decrease in the depression and/or anxiety score of one standard deviation is associated with an increase in engaging in paid rather than unpaid work of 5% points. Although column (4) is shown for completeness' sake, we do not comment on it as it is not sufficiently powered: only 17 women switched from not working to unpaid worker status or vice versa between baseline and follow-up (Table 6, Panel B).

Women who switch out of probable depression or anxiety are also 15.3% points more likely to engage in a secondary activity (31.9% increase in column 5) and have a slightly higher productivity (total earnings per hour in column 6). Finally, and not surprisingly, we find that local functioning impairment is negatively correlated with labor force participation: women who face one standard deviation increase in LFI index score (where a higher score indicates worse functioning) are 10.5% points less likely to work, 11% points less likely to engage in paid work, and 5.6% points less likely to engage in a secondary activity. They also had significantly lower earnings in the last 7 days (by about 14.9%), which seems suggestively driven by a non-significant decrease in the number of hours worked.

In Table 8, following Eq (1), we look at the same set of regressions as in Table 7, this time controlling for the LFI index. By doing so, we are attempting to estimate the correlation between work and mental health net of the impact that improved PTSD or depression might have on local functioning.

**Table 8. Panel regression of women's work on mental health outcomes, controlling for LFI index.**

|  | (1) | (2) | (3) | (4) | (5) | (6) | (7) | (8) |
|---|---|---|---|---|---|---|---|---|
|  | Worked in the last 7 days | Paid work last 7d (= 1 if worked with earnings; = 0 if worked w/o earnings) | Paid work last 7d (= 1 if worked with earnings; = 0 if not worked) | Unpaid work last 7d (= 1 if worked with no earnings; = 0 if not worked) | Respondent has a secondary activity | Total earnings per hour last 7d | Total earnings generated last 7d | Total hours worked last 7d |
| PTSD checklist score | 0.0308** | -0.0279 | 0.0304* | 0.01000 | -0.0213 | -0.00613 | 0.372 | 1.731** |
|  | (0.0147) | (0.0233) | (0.0168) | (0.0678) | (0.0269) | (0.0101) | (0.258) | (0.873) |
| Probable PTSD | 0.0150 | -0.0688 | 0.00648 | -0.0129 | -0.0579 | -0.0108 | 0.790 | 2.971* |
|  | (0.0282) | (0.0442) | (0.0325) | (0.131) | (0.0516) | (0.0189) | (0.496) | (1.648) |
| HSCL-25 score | 0.0292* | -0.0455* | 0.0342* | -0.0347 | -0.0265 | -0.00685 | 0.221 | 0.786 |
|  | (0.0167) | (0.0260) | (0.0189) | (0.0754) | (0.0305) | (0.0114) | (0.294) | (0.991) |
| Probable dep or anx | 0.0509 | -0.0710 | 0.0446 | 0.0509 | -0.130** | -0.0347* | 0.0519 | 2.888 |
|  | (0.0315) | (0.0495) | (0.0363) | (0.142) | (0.0579) | (0.0210) | (0.561) | (1.858) |
| Observations | 985 | 927 | 839 | 244 | 936 | 849 | 935 | 885 |
| Individual Fixed Effects | Y | Y | Y | Y | Y | Y | Y | Y |
| Survey round dummy | Y | Y | Y | Y | Y | Y | Y | Y |
| Controlling for LFI index | Y | Y | Y | Y | Y | Y | Y | Y |
| Baseline Mean | 0.881 | 0.806 | 0.856 | 0.586 | 0.479 | 0.130 | 3.446 | 32.11 |

Notes: Each cell shows the estimation result for the regression of one labor outcome on one mental health explanatory variable. The regressions reported in the same column have the same sample size given in the observations row. Panel regressions are using baseline and follow-up observations for the no-therapy group, and controls for individual fixed effects ("Y" stands for yes for controlling for individual fixed effects and survey round dummy). Mental health outcomes and LFI index are standardized. Earning values are winsorized at the top 1% level. Standard errors in parentheses.

* $p < 0.1$

** $p < 0.05$

*** $p < 0.01$.

Introducing the LFI index reveals a positive correlation between work and mental health symptoms. For women with similar levels of local functioning, working is associated with higher levels of both PTSD score and HSCL-25 score: both a one standard deviation increase in PTSD score and in HSCL-25 score are associated with a 3 percentage point increase in the likelihood of working compared to not working (column 1). This seems entirely driven by paid work which exhibits similar correlations with the PTSD checklist and HSCL-25 scores (column 3). For women who worked at baseline and follow-up, controlling for the LFI index weakens the negative associations between their mental health and doing paid rather than unpaid work, which is no longer significantly associated with probable PTSD and probable depression or anxiety, even though the magnitude does not change much (column 2). At the same time, a decrease in one standard deviation of the HSCL-25 score still increases the probability of switching from unpaid to paid work between baseline and follow-up by 4.5 percentage points.

The other robust relationships are the ones between probable depression or anxiety and secondary activity and productivity. Even after accounting for local functioning impairment, women who switch out of probable depression or anxiety at follow-up are more likely to engage in a secondary activity and have higher productivity. Local functioning being held constant, women who switched from non-probable PTSD to probable PTSD at follow-up also worked around three hours more in the last 7 days.

Based on the suggestive evidence in Table 8 where working is associated with an increase in PTSD symptoms, we explore which cluster of PTSD symptoms may be driving these effects. Zooming in on symptom clusters shows that the PTSD hyperarousal symptoms, when not controlling for LFI (Table 9, column 2) are negatively associated with doing paid vs. unpaid work. A decrease in PTSD hyperarousal symptoms of one standard deviation is associated with an increase in the likelihood of switching from unpaid to paid word (among women working at both baseline and follow-up) of 3.9% points, or 4.9%.

In Table 10, we introduce LFI as an additional control. We find that the positive correlations between PTSD and labor force participation found in Table 8 are driven by PTSD's re-experiencing and hyperarousal symptoms. Similar significant relationships hold between re-experiencing and hyperarousal symptoms and the likelihood of doing paid work compared to not working. For re-experiencing symptoms, we also find a positive association with the number of hours worked. Furthermore, we now see that an increase in earnings is also associated with an increase in re-experiencing symptoms. Women who see a one standard deviation increase in re-experiencing symptoms between baseline and follow-up also work 2.1 hours more, which translates into a USD 0.47 (or 13.6%) increase in total earnings in the last 7 days.

## Type of work and mental health

In this section we examine how the type of work that women do correlates with their mental health outcomes. In Table 11 we show results from cross-sectional regressions, where each cell represents the result of a regression of the employment variable on a mental health outcome. In all regressions reported in this table, the subsample of women in unpaid worker status is excluded.

Women with worse PTSD symptoms are less likely to be self-employed in their main activity and more likely to be engaged in farming. An increase in one standard deviation of the PTSD checklist score is associated with a 2.7% point decrease in self-employment (column 1) and a 2.8% point increase in farming (column 2). The relationship between self-employment and PTSD is driven by women who are in off-farm activities, that is, women with a lower PTSD score are more likely to run a non-agricultural business (column 4). The relationships

**Table 9. Panel regression of women's work on PTSD symptom clusters.**

| | (1) | (2) | (3) | (4) | (5) | (6) | (7) | (8) |
|---|---|---|---|---|---|---|---|---|
| | Worked in the last 7 days | Paid work last 7d (= 1 if worked with earnings; = 0 if worked w/o earnings) | Paid work last 7d (= 1 if worked with earnings; = 0 if not worked) | Unpaid work last 7d (= 1 if worked with no earnings; = 0 if not worked) | Respondent has a secondary activity | Total earnings per hour last 7d | Total earnings generated last 7d | Total hours worked last 7d |
| PTSD score—re-experiencing | 0.00845 | -0.0320 | 0.0114 | -0.00759 | -0.0372 | -0.000582 | 0.335 | 1.563** |
| | (0.0145) | (0.0211) | (0.0169) | (0.0687) | (0.0249) | (0.00918) | (0.240) | (0.787) |
| PTSD score—avoidance | -0.000543 | -0.0160 | -0.00429 | 0.0586 | -0.0127 | -0.0141 | 0.126 | 0.594 |
| | (0.0146) | (0.0213) | (0.0164) | (0.0791) | (0.0251) | (0.00891) | (0.242) | (0.795) |
| PTSD score—hyperarousal | -0.000166 | -0.0394* | 0.00774 | -0.0406 | -0.0423 | 0.000619 | 0.115 | 0.844 |
| | (0.0153) | (0.0223) | (0.0186) | (0.0634) | (0.0264) | (0.00981) | (0.254) | (0.838) |
| Observations | 985 | 927 | 839 | 244 | 936 | 849 | 935 | 885 |
| Individual Fixed Effects | Y | Y | Y | Y | Y | Y | Y | Y |
| Survey round dummy | Y | Y | Y | Y | Y | Y | Y | Y |
| Baseline Mean | 0.881 | 0.806 | 0.856 | 0.586 | 0.479 | 0.130 | 3.446 | 32.11 |

Notes: Each cell shows the estimation result for the regression of one labor outcome on one mental health explanatory variable. The regressions reported in the same column have the same sample size given in the observations row. Panel regressions are using baseline and follow-up observations for the no-therapy group, and controls for individual fixed effects ("Y" stands for yes for controlling for individual fixed effects and survey round dummy). Mental health outcomes are standardized. Earning values are winsorized at the top 1% level. Standard errors in parentheses.

* $p<0.1$

** $p<0.05$

*** $p<0.01$.

are similar when looking at the anxiety and depression score. Women with higher levels of anxiety or depression are significantly less likely (3.4% points) to be self-employed and more likely to be engaged in farming (2.6% points). Women with higher HSCL-25 scores are also significantly more likely to be wage workers, in particular on-farm wage workers (3% points). The increasing relationship between farming (and on-farm wage work) and mental health disorders is only found when looking at primary activities. Results for farming and wage work as primary or secondary activities are available upon request. We do not see significant correlations between probable PTSD or probable depression and/or anxiety and employment status. There is a significant correlation between local functioning and the type of work women do: one standard deviation increase in the LFI index is associated with a 3.3% points increase in the likelihood of working on farm for wages for women engaged in paid work in our sample.

## Discussion

Unlike previous research on the topic, we do not find that increases in symptoms of PTSD, and/or depression and anxiety, reduce women's overall likelihood of engaging in work, whether paid or unpaid. However, we do find that for a given woman, increases in depression and/or anxiety symptoms correlate with a lower likelihood of being employed in paid activities compared to unpaid work. Increased likelihood of probable depression and/or anxiety are also associated with reduced likelihood of secondary income generating activities and with reduced hourly earnings. In addition to symptom severity, functional impairment is a key criterion for diagnosis of a mental disorder [34]. This hampering of day to day social and occupational

**Table 10. Panel regression of women's work on PTSD symptom clusters, controlling for LFI index.**

| | (1) | (2) | (3) | (4) | (5) | (6) | (7) | (8) |
|---|---|---|---|---|---|---|---|---|
| | Worked in the last 7 days | Paid work last 7d (= 1 if worked with earnings; = 0 if worked w/o earnings) | Paid work last 7d (= 1 if worked with earnings; = 0 if not worked) | Unpaid work last 7d (= 1 if worked with no earnings; = 0 if not worked) | Respondent has a secondary activity | Total earnings per hour last 7d | Total earnings generated last 7d | Total hours worked last 7d |
| PTSD score–re-experiencing | 0.0323** | -0.0271 | 0.0335** | 0.0178 | -0.0266 | 0.000910 | 0.468* | 2.101** |
| | (0.0139) | (0.0221) | (0.0160) | (0.0639) | (0.0255) | (0.00958) | (0.244) | (0.820) |
| PTSD score—avoidance | 0.0173 | -0.0114 | 0.0145 | 0.000308 | -0.00315 | -0.0136 | 0.221 | 0.917 |
| | (0.0139) | (0.0218) | (0.0156) | (0.0760) | (0.0254) | (0.00915) | (0.245) | (0.815) |
| PTSD score—hyperarousal | 0.0300** | -0.0347 | 0.0343* | 0.00794 | -0.0292 | 0.00256 | 0.272 | 1.411 |
| | (0.0149) | (0.0234) | (0.0177) | (0.0610) | (0.0274) | (0.0103) | (0.262) | (0.884) |
| Observations | 985 | 927 | 839 | 244 | 936 | 849 | 935 | 885 |
| Individual Fixed Effects | Y | Y | Y | Y | Y | Y | Y | Y |
| Survey round dummy | Y | Y | Y | Y | Y | Y | Y | Y |
| Baseline Mean | 0.881 | 0.806 | 0.856 | 0.586 | 0.479 | 0.130 | 3.446 | 32.11 |

Notes: Each cell shows the estimation result for the regression of one labor outcome on one mental health explanatory variable. The regressions reported in the same column have the same sample size given in the observations row. Panel regressions are using baseline and follow-up observations for the no-therapy group, and controls for individual fixed effects ("Y" stands for yes for controlling for individual fixed effects and survey round dummy). Mental health outcomes and LFI index are standardized. Earning values are winsorized at the top 1% level. Standard errors in parentheses.

\* $p<0.1$

\*\* $p<0.05$

\*\*\* $p<0.01$.

functioning is a key pathway through which mental illness impacts employment and productivity. When independently looking at the correlation between functional impairment and employment, we find that as levels of daily functioning worsen for a given woman, she is less likely to be working altogether, less likely to be carrying out paid or unpaid work, less likely to be engaged in secondary income generating activities, and more likely to have reduced earnings.

We hypothesize that local functioning is the main pathway through which mental health affects work and hence look at the association between employment and mental health net of the impacts of functioning on work. This helps us tease out the ways in which work may be impacting mental health. Upon controlling for this pathway, several associations appear. Importantly, we find a positive association between depression and/or anxiety and PTSD symptom severity and working, especially paid employment. We also find that an increase in number of hours worked (paid or unpaid) in the past week for a given woman is associated with an increase in PTSD symptom severity and probable PTSD. Our results provide suggestive evidence that, for this population that is already suffering from high levels of PTSD symptoms at baseline, working or spending more time working is associated with worsening mental health symptoms once we account for the impact of improved daily functioning on work. One could argue that a common cause such as economic distress is driving the link between increased work and worsened mental health. Given that we are using longitudinal data, this would need to happen through a worsening of economic distress at the individual level leading to both increased working and worsened mental health. However, having a secondary income

**Table 11. Cross-section regressions of women's employment sector and status and mental health outcomes.**

|  | (1) | (2) | (3) | (4) | (5) | (6) |
|---|---|---|---|---|---|---|
|  | Self-employed (= 0 if Wage worker) | Farming (= 0 if off-farm) | On-farm and self-employed | Off-farm and self-employed | On-farm and wage worker | Off-farm and wage worker |
| PTSD checklist score | -0.0272* | 0.0278** | 0.00304 | -0.0302** | 0.0247 | 0.00249 |
|  | (0.0161) | (0.0141) | (0.0166) | (0.0128) | (0.0152) | (0.00985) |
| Probable PTSD | -0.0130 | 0.0191 | 0.0252 | -0.0382 | -0.00605 | 0.0191 |
|  | (0.0348) | (0.0293) | (0.0351) | (0.0279) | (0.0321) | (0.0198) |
| HSCL-25 score | -0.0343** | 0.0256* | -0.00490 | -0.0294** | 0.0305** | 0.00382 |
|  | (0.0159) | (0.0139) | (0.0167) | (0.0124) | (0.0152) | (0.0103) |
| Probable dep or anx | 2.01e-05 | 0.00702 | 0.00854 | -0.00852 | -0.00152 | 0.00150 |
|  | (0.0382) | (0.0333) | (0.0395) | (0.0315) | (0.0348) | (0.0239) |
| LFI index | -0.0253 | 0.00701 | -0.0265 | 0.00120 | 0.0335** | -0.00821 |
|  | (0.0166) | (0.0146) | (0.0166) | (0.0123) | (0.0159) | (0.0110) |
| Observations | 883 | 883 | 883 | 883 | 883 | 883 |
| Baseline Mean | 0.563 | 0.736 | 0.404 | 0.159 | 0.332 | 0.105 |

Notes: Each cell shows the estimation result for the regression of one labor outcome on one mental health explanatory variable. The number of observations in all the regressions included in this table is 883. In all regressions the subsample of women in paid worker status (self-employed or wage worker) is used, women in unpaid worker status are excluded. Mental health outcomes and LFI index are standardized. Control variables included: age; marital status; and schooling of the respondent; dummy for household head; household size; durable and livestock ownership pca scores; total amount of savings in the last 12 months in USD (winsorized at 2.5%); dummy for access to funds; dummy for food insecurity; and strata dummies. Robust standard errors in parentheses.

\* $p<0.1$

\*\* $p<0.05$

\*\*\* $p<0.01$.

generating activity, engaging in paid vs. unpaid work, and increased hourly earnings, all indicators of reduced economic distress, continue to be associated with reduced likelihood of probable depression and/or anxiety even after accounting for LFI, while paid compared to no work is what is associated with worsening of depression and/or anxiety symptoms. The associations we see are supported by another study on the impacts of a business training, a cash grant and support to set up a business in Uganda: Green et al. [18] indicate that despite the positive impacts of economic well-being on mental health, their study does not see positive effects on depression due to the gains being offset by the stress of entrepreneurial activities.

Our consistent finding that an increase in hourly earnings reduces depression and/or anxiety independent of functioning support the general finding in the literature that cash transfers can reduce symptoms of depression. This was also seen amongst conflict affected populations in a recent study by Glass et al. [16] who found that asset transfers improved symptoms of depression. In a recent qualitative study, some protracted refugees in Jordan also reported that cash transfers reduce both stress and anxiety through providing financial security and ability to pay rent [36, 37].

For PTSD, there is little evidence that earnings or cash transfers can alleviate symptoms. Upon accounting for LFI a positive association between PTSD symptoms and paid vs. no work emerges. Due to the worsening of PTSD symptoms when women work, after accounting for LFI, we carried out an exploratory analysis that looks at symptom clusters of PTSD and their association with work status, hours worked, and earnings to unpack which sub-set of PTSD symptoms are driving the positive association between PTSD and employment. In the absence of controlling for LFI, reduction in hyperarousal alone appears to drive the association

between paid work and reduced PTSD. The association between indicators of hyperarousal and re-experiencing both may be driving the overall association between probable PTSD and overall working as well as paid employment after LFI is accounted for. Hyperarousal symptoms have been previously considered to be a strong indicator of poor physical functioning and greater disability [38–40]. The association between greater number of hours worked and PTSD is also driven by increased symptoms of re-experiencing. While hyperarousal symptoms are constant, re-experiencing symptoms result from exposure to triggers [41] therefore strengthening the argument that this positive association between work and PTSD symptoms is potentially driven by increased exposure to triggers. This is further reinforced by the positive association between greater earnings and hours at work and re-experiencing symptoms. In some studies re-experiencing is associated with exposure to physical or sexual interpersonal violence by a non-intimate partner [42]. The fact that an increase in symptoms related to re-experiencing is associated with increased hours worked and increased total earnings could be indicative of employment putting women at greater risk of violence and thereby worsening their mental health through this. While our data does not permit us to explore this any further, future research should pay a close attention to any causal negative relationship going from working to mental health symptoms.

We also attempt to unpack the ways in which sector of work could be negatively associated with mental health. In line with nationally representative surveys in the DRC such as DHS, we find that women workers in our sample are mostly engaged with agriculture/farming. Women with worse PTSD symptoms (scores) and depression and/or anxiety are more likely to be engaged in farming or wage work and less likely to be engaged in self-employment (especially off-farm self-employment). This is in line with much of the literature in the general population, where farm-based employment is associated with a stronger likelihood of mental disorders [11]. Factors such as pesticide use, financial pressures and uncertainties, climate change and poor health due to occupational injury appear to contribute to this [11]. Additionally, the worsening of mental health due to farm-work and wage work may also be directly due to the quality of employment this sector provides. This is in line with Butterworth et al. [12] and Rönnblad et al. [43] who note that poor quality and precarious employment can have negative impacts on mental health, that are no different from the effects of unemployment. Our data also indicate that in addition to the choice of occupation being related to mental health, functional impairment may also drive the choice of sector as we find that those with greater functional impairment are more likely to be employed on-farm and in wage work.

There are some limitations to this study. Firstly, our data does not enable us to make statements about the direction of causality between mental health and employment outcomes. With the data at hand, we can only make hypotheses as to which is affecting which, but we cannot say for sure. To the extent that local functioning is a primary pathway through which mental health affects labor market outcomes, we assume that any remaining correlation between the two, once local functioning is controlled for (as well as other key common confounders such as wealth and education), stems largely from employment affecting mental health. However, assuming that only local functioning explains the mental health to employment effect is a strong assumption. Second, the fixed effect model used in the study does not eliminate the risk of bias posed by some omitted idiosyncratic time-varying variables correlated with both mental health and women's labor market outcomes. However, the model does limit the potential sources of biases, in comparison to a standard Ordinary Least Squares model, in which a correlation between any unobserved variable (both time-variant and invariant) would result in biased estimates. Third, we lack data on the context and setting of the work the women do, for farming in particular. Knowing more about the location of the plots in relation to their home or whether they work with other people on the plot would help understand why farming in

particular seems to be associated with PTSD symptoms, especially hyperarousal, and depression and anxiety. Fourth, we do not have information on the sources of their (likely multiple) traumas. Ethical considerations prevented the research team from collecting those data. The type of trauma and the setting in which it occurred may also contribute to explaining the results around farming and mental health. For example, if a traumatic event took place near a field in a situation of isolation, then working in those circumstances could trigger symptoms. Finally, as we run a large number of regressions to uncover patterns between labor market outcome and mental health indicators, we acknowledge the likelihood of false rejection of null hypotheses due to multiple hypotheses testing. However, the purpose of this study is mainly descriptive and exploratory, so we do not correct for multiple inference, following recommendations by Bender and Lange (2001).

Despite these limitations, this study makes an important contribution to the existing literature on the relationship between mental health and labor market outcomes for women in conflict affected settings. Most studies use data from developed countries, a setting very different from the DRC, which is in a protracted conflict situation and has extremely high poverty rates with very few or no safety nets. In this context, survival necessity requires that almost everyone be engaged in some economic activity, be it farming, self-employment, or to a lesser extent, wage work. The relationship between one's mental health and participation in the labor market must then be examined with a different lens than in a non-conflict, less resource-constrained setting. In addition, the availability of panel data, with two observation points for each individual, enables us to control for a host of unobservable characteristics that would otherwise limit our ability to draw conclusions regarding the link between mental health and employment. Using panel data, we can examine how a change in mental health is associated with a change in employment status, net of the effect of confounding, unobservable factors.

## Conclusion

There is limited evidence on the association between mental health and labor market outcomes among vulnerable populations in post-conflict settings. Using original data from eastern DRC, this study finds that changes in mental health symptoms among women survivors of conflict appear not to be associated with changes in overall labor force participation, however amongst those working engagement in paid work is associated with improved mental health. Improved mental health is also found to be associated with a higher likelihood of engaging in a secondary income generating activity and with higher productivity. Accounting for local functioning impairment as a possible mechanism in the pathway from mental health to work reveals that working, in particular carrying out paid work, is associated with worse mental health symptoms. This may suggest that while improvements in local functioning are positively associated with working, working itself may be associated with worsened mental health in this setting. An increase in the number of hours worked seems to increase women's PTSD levels, specifically re-experiencing symptoms. We find suggestive evidence that on-farm wage work is associated with worse mental health and may even deteriorate mental health.

More research is needed to refine our understanding of the effect of work and different types of work on the mental health of individuals suffering from high levels of PTSD, depression, and anxiety. The findings of this study point to the need to pay close attention to the mental health of beneficiaries when designing and implementing social and economic empowerment programs for women survivors of violence. In post-conflict settings, where mental health disorders may be severe and prevalent, programming should take into account that some livelihood interventions targeted at specific sectors might actually increase mental health disorders' symptoms. This could take the form of a robust monitoring system of women

beneficiaries' mental and physical wellbeing. Regular data collections and the involvement of trained case managers in such activities may help detect adverse effects of economic inclusion programs. In addition, treatment for mental health disorders could be included as a component of economic programming.

## Supporting information

**S1 Table. Balance tests between the therapy and no-therapy groups at baseline.**
(DOCX)

**S1 File. Minimal baseline dataset.**
(DTA)

**S2 File. Minimal panel dataset.**
(DTA)

**S3 File. Minimal merged dataset.**
(DTA)

**S4 File. PLOS' questionnaire on inclusivity in global research.**
(DOCX)

## Acknowledgments

This paper is a product of the Africa Gender Innovation Lab, Office of the Chief Economist, Africa Region. The findings, interpretations, and conclusions expressed in this work do not necessarily reflect the views of The World Bank Group, its Board of Executive Directors, or the governments they represent. The authors would like to thank Moussa Sawadogo for excellent fieldwork and the Fonds Social de la République Démocratique du Congo for a fruitful collaboration on this research project, and Pia Peeters, Patricia Fernandes and Verena Phipps for their support throughout this project.

## Author Contributions

**Conceptualization:** Julia Vaillant, Naira Kalra, Alev Gurbuz Cuneo, Léa Rouanet.

**Formal analysis:** Julia Vaillant, Naira Kalra, Alev Gurbuz Cuneo, Léa Rouanet.

**Methodology:** Julia Vaillant, Naira Kalra, Alev Gurbuz Cuneo, Léa Rouanet.

**Writing – original draft:** Julia Vaillant, Naira Kalra, Alev Gurbuz Cuneo, Léa Rouanet.

**Writing – review & editing:** Julia Vaillant, Naira Kalra, Alev Gurbuz Cuneo, Léa Rouanet.

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
