## [Decision Letter · Decision Letter 0]

2 Jan 2023

PONE-D-22-22896Losing ground in the field: An exploratory analysis of the relationship between work and mental health amongst women in conflict affected Democratic Republic of the CongoPLOS ONE

Dear Dr. Kalra,

Thank you for submitting your manuscript to PLOS ONE. After careful consideration, we feel that it has merit but does not fully meet PLOS ONE’s publication criteria as it currently stands. Therefore, we invite you to submit a revised version of the manuscript that addresses the points raised during the review process.

We look forward to receiving your revised manuscript.

Kind regards,

Tae-Young Pak, Ph.D.

Academic Editor

PLOS ONE

Journal Requirements:

Reviewers' comments:

Reviewer's Responses to Questions

**Comments to the Author**

1. Is the manuscript technically sound, and do the data support the conclusions?

Reviewer #1: Partly

Reviewer #2: Partly

Reviewer #3: Yes

2. Has the statistical analysis been performed appropriately and rigorously? 

Reviewer #1: No

Reviewer #2: Yes

Reviewer #3: Yes

3. Have the authors made all data underlying the findings in their manuscript fully available?

Reviewer #1: Yes

Reviewer #2: Yes

Reviewer #3: Yes

4. Is the manuscript presented in an intelligible fashion and written in standard English?

Reviewer #1: Yes

Reviewer #2: Yes

Reviewer #3: Yes

5. Review Comments to the Author

Reviewer #1: I am well aware of how challenging it is to do this kind of research in the settings the authors are working in. Nevertheless, I have some serious reservations about the presentation of the data, and the analyses they have performed. A minor point first.

The authors report a substantial participation and follow up rate- how were the responses obtained- was it a paper and pencil process, or an electronic submission, or some other procedure. and what was the refusal to participate rate?

But my major reservations are twofold.

Firstly, a large number of interactions have been reported using various component scores-given the number of analyses involved, would a number have been significant by chance alone?

And more importantly, the panel procedures that have been used do not allow for interaction terms. The analyses of impairment as a mediator between mental health and employment has been dealt with by analysing the mental health/employment relationship at various fixed level of impairment. I don't think this is the correct way to do that. I think what is needed is the development of a multivariate regression model with interaction terms to look at the relative weight of the various factors- I don't think a panel approach is suitable for that, and I suggest the authors consult a biostatistical consultant and discuss creating a more comprehensive mathematical model that would be more flexible, and easier to understand in relation to the relative importance of the variable that have been studied. The sample size is quite large enough for model building.

Reviewer #2: The study covers a research topic which has been well studied and published in the recent years. The mental health of women has been studied in context to various factors in the region (with relation to conflict) Examples include:

Mitonga-Monga J, Mayer CH. Sense of coherence, burnout, and work engagement: The moderating effect of coping in the Democratic Republic of Congo. International Journal of Environmental Research and Public Health. 2020 Jan;17(11):4127.

Vivalya BM, Bin Kitoko GM, Nzanzu AK, Vagheni MM, Masuka RK, Mugizi W, Ashaba S. Affective and psychotic disorders in war-torn eastern part of the Democratic Republic of the Congo: a cross-sectional study. Psychiatry journal. 2020 Jul 24;2020.

Dossa NI, Zunzunegui MV, Hatem M, Fraser WD. Mental health disorders among women victims of conflict-related sexual violence in the Democratic Republic of Congo. Journal of Interpersonal Violence. 2015 Aug;30(13):2199-220.

Johnson K, Scott J, Rughita B, Kisielewski M, Asher J, Ong R, Lawry L. Association of sexual violence and human rights violations with physical and mental health in territories of the Eastern Democratic Republic of the Congo. Jama. 2010 Aug 4;304(5):553-62.

The study adds very little to existing knowledge base and may be of limited interest to wider readership. Perhaps a local Journal might be interested in the study.

Reviewer #3: The manuscript was well written and the subject is very interesting. With the context of the Democratic Republic of Congo, these data will help improve the care of women who have suffered violence. This manuscript was well written.

6. PLOS authors have the option to publish the peer review history of their article (what does this mean?). If published, this will include your full peer review and any attached files.

Reviewer #1: No

Reviewer #2: No

Reviewer #3: **Yes: **Aketi Paizanos Loukia

---

## [Author Response · Author response to Decision Letter 0]

15 Mar 2023

Response to the editor and reviewer comments

Response to Editor’s comments:

Author response: Thank you for flagging these. We have reviewed the style template and fixed any style requirements we had missed. 

Author response: Attached as Supporting Information.

Author response: Thank you for the guidance. We have added the following text to the methods section of the manuscript “Written informed consent was collected at study registration, baseline and follow-up data collection.”

4. We note that the grant information you provided in the ‘Funding Information’ and ‘Financial Disclosure’ sections do not match. When you resubmit, please ensure that you provide the correct grant numbers for the awards you received for your study in the ‘Funding Information’ section.

Author response: There was no specific grant for this work and hence no grant number or awards to specify. However, we would like to continue to acknowledge the World Bank Group’s Nordic Trust Fund, and the Swiss Development Cooperation for financial support that was directed towards this work.

Author response: We have attached three data files as Supporting Information. We would like to change this statement to say “All relevant data are within the paper and its Supporting Information files”. This has also been added in the cover letter.

Response to reviewer comments:

Reviewer #1: I am well aware of how challenging it is to do this kind of research in the settings the authors are working in. Nevertheless, I have some serious reservations about the presentation of the data, and the analyses they have performed. A minor point first.

The authors report a substantial participation and follow up rate- how were the responses obtained- was it a paper and pencil process, or an electronic submission, or some other procedure. and what was the refusal to participate rate?

Author response: The data were collected electronically, using tablets and the SurveyCTO software. Consent to participate was recorded at two points in the study: i) registration stage: the counselors offered participants the option of being part of the study or not. Signed consent was obtained at that point for individuals who consented to participate in the study. We do not have data on the refusal rates at that stage as those who did not consent were not recorded in a registry; ii) data collection stage: enumerators collected consent at baseline and follow-up. One refusal to participate was recorded at follow-up. Other reasons for attrition were having moved out of the area (13 cases) and not being found by enumerators (10 cases). 

The following was added to the paper: “Main reasons for attrition were not being found (10 cases) and having moved out of the area (13 cases). One refusal case was recorded at follow-up.”

But my major reservations are twofold.

Firstly, a large number of interactions have been reported using various component scores-given the number of analyses involved, would a number have been significant by chance alone?

Author response: Thank you for this comment. One of the motivations for the multiple mental health measures included in the analysis (as opposed to a synthetic index) is that the literature on the link between economic outcomes and PTSD is very scarce (see introduction, line 68). We did not consider the option of reducing the number of mental health measures (to reduce the number of inferences made) because one of the main contributions that this study makes is to include granular validated mental health indicators. The literature on the link between PTSD and economic outcomes is particularly scarce, therefore showing how labor market outcomes are differentially correlated with depression/anxiety and PTSD seems a contribution of value to the literature. 

As the study is exploratory in nature and attempts to uncover patterns of correlations between labor outcomes and mental health indicators, we have not corrected for multiple hypothesis testing (MHT) in this paper. It is true that to produce the analysis and draw conclusions, a large number of regressions had to be run. We follow the views of Bender and Lange (2001), who indicate that when the purpose of a study is mainly descriptive and not to draw conclusions for decision-making purposes, correcting for multiple inference is not necessary. Our results are clearly labelled as exploratory and would have to be confirmed by a study designed specifically to confirm them. 

Following the reviewer comment, we did calculate sharpened False Discovery Rate (FDR) q-values, following Anderson (2008). The FDR is the expected proportion of rejections that are type I errors (false rejections). Q-values were calculated for the main tables of the paper (Tables 7 and 8) on the following set of outcome variables: worked in the past 7 days, worked for pay (vs unpaid work), hourly earnings. Other outcomes in the tables are considered secondary to the analysis or are components of the main outcome. For example, we do not include hours of work or total earnings in the calculation of sharpened q-values, because they are components of hourly earnings. Coefficients that are significant at the 1% level remain significant when adjusting for multiple inference, whereas coefficients significant at the 5% and 10% level now have q-values above 10%. This is expected with the high number of regressions that we ran and coefficients being significant at the 5% and 10% levels. However, regressions of several labor market outcomes on a number of mental health indicators do show correlations that go in the same direction and are consistent with each other. Consequently, a clear pattern is uncovered in Tables 7 and 8, even though some coefficients are weakly statistically significant. Our level of statistical power may be too weak to demonstrate a relationship between labor market outcome and mental health indicators with absolute statistical certainty, but the empirical patterns uncovered in this analysis do provide some suggestive evidence. For example, in Table 8, when we control for the LFI variable, we clearly see a pattern of positive correlation between working and mental health indicators, and some of these coefficients are significant at the 10% and 5% level. It is reasonable to interpret these results as a real pattern and not the result of coefficients being significant at random due to the large number of regressions being run. Q-values are available upon request. 

We added the following to the limitations paragraph in the paper: “Finally, as we run a large number of regressions to uncover patterns between labor market outcome and mental health indicators, we acknowledge the likelihood of false rejection of null hypotheses due to multiple hypotheses testing. However, the purpose of this study is mainly descriptive and exploratory, so we do not correct for multiple inference, following recommendations by Bender and Lange (2001).”

Bender, R., & Lange, S. 2001. Adjusting for multiple testing—when and how? Journal of clinical epidemiology, 54(4), 343-349.

Anderson, M. L. (2008). Multiple inference and gender differences in the effects of early intervention: A reevaluation of the Abecedarian, Perry Preschool, and Early Training Projects. Journal of the American Statistical Association, 103(484), 1481-1495.

And more importantly, the panel procedures that have been used do not allow for interaction terms. The analyses of impairment as a mediator between mental health and employment has been dealt with by analysing the mental health/employment relationship at various fixed level of impairment. I don't think this is the correct way to do that. I think what is needed is the development of a multivariate regression model with interaction terms to look at the relative weight of the various factors- I don't think a panel approach is suitable for that, and I suggest the authors consult a biostatistical consultant and discuss creating a more comprehensive mathematical model that would be more flexible, and easier to understand in relation to the relative importance of the variable that have been studied. The sample size is quite large enough for model building.

Author response: Thank you for this comment and suggestion. To clarify, we are not per se analyzing the role of impairment as a mediator between mental health in Tables 8 and 10. We apologize to the reviewer, as we were using the word ‘mediator’ a couple of times in the previous version of the manuscript, which created confusion. This is now fixed in the current version of the paper. If we were to conduct mediation analysis, we would aim to assess empirically the relationship between local functioning impairment and women’s work, following changes in mental health at the individual level. Concretely, we would indeed interact the LFI index with the mental health outcome, and report both the LFI coefficients, and the interaction coefficients in Tables 8 and 10. In the setting of this study, we are not able to empirically assess the impact of an enhancement of local functioning happening through improved mental health on women’s work with mediation analysis. Indeed, according to Heckman and Pinto (2015), “a fundamental problem of mediation analysis is that even though we might observe experimental variation in some inputs and outputs, the relationship between inputs and outputs might be confounded by unobserved variables. There may exist relevant unmeasured inputs changed by the experiment that impact outputs. If unmeasured inputs are not statistically independent of measured ones, then the observed empirical relation between measured inputs and outputs might be due to the confounding effect of experimentally induced changes in unmeasured inputs. In this case, treatment effects on outputs can be wrongly attributed to the enhancement of measured inputs instead of experimentally induced increase in unmeasured inputs.” By controlling for local functioning impairment in Tables 8 and 10, we aim to estimate the association between labor market and mental health measures net of the impact a respondents’ level of PTSD or depression and/or anxiety might have on their functioning. We then rely on existing literature (Harnois et al., 2000 and Ridley et al., 2020) to hypothesize that local functioning is a key pathway through which mental health affects work. Consequently, controlling for this pathway helps us tease out the ways in which work may be impacting mental health. 

In addition, we make use of a panel approach to use a fixed effects regression model. Such models are widely used for causal inference with longitudinal or panel data in order to adjust for unobserved, unit-specific and time-invariant confounders when estimating causal effects from observational data (e.g., Angrist and Pischke 2009). The main benefit of fixed effects estimations is that the potential sources of biases in the estimations are limited in comparison to classical OLS models. In the case of OLS models, a correlation between any unobserved variable and the outcome or the treatment variable of interest results in a biased estimate of the treatment effect. In contrast, fixed effects models limit the sources of bias to time-varying variables that correlate with the treatment as well as with the outcome over time, which is far more achievable than the strong exogeneity assumption of OLS models. We cannot rule out that some key omitted variables in the relationship between mental health and economic participation are not time invariant. As a result, in our study, estimates could be biased by idiosyncratic time-varying variables that would correlate both with mental health and women’s work. 

We added the following to the text, in the limitations section: “Second, the fixed effect model used in the study does not eliminate the risk of bias posed by some omitted idiosyncratic time-varying variables correlated with both mental health and women’s labor market outcomes. However, the model does limit the potential sources of biases, in comparison to a standard Ordinary Least Squares model, in which a correlation between any unobserved variable (both time-variant and invariant) would result in biased estimates.”

Angrist, Joshua D., and Jorn-Steffen Pischke. 2009. Mostly Harmless Econometrics: An Empiricist’s Companion. Princeton, NJ: Princeton University Press

Harnois G, Gabriel P. Mental health and work: impact, issues and good practices. Geneva: World Health Organization, International Labour Organisation; 2000.

Heckman J, Pinto R. Econometric Mediation Analyses: Identifying the Sources of Treatment Effects from Experimentally Estimated Production Technologies with Unmeasured and Mismeasured Inputs. Econom Rev. 2015;34(1-2):6-31. doi: 10.1080/07474938.2014.944466. PMID: 25400327; PMCID: PMC4228489.

Ridley M, Rao G, Schilback F, Patel V. Povret, depression, and anxiety: Causal evidence and mechanisms. Science. 2020; 370(5522).

Reviewer #2: The study covers a research topic which has been well studied and published in the recent years. The mental health of women has been studied in context to various factors in the region (with relation to conflict) Examples include:

Mitonga-Monga J, Mayer CH. Sense of coherence, burnout, and work engagement: The moderating effect of coping in the Democratic Republic of Congo. International Journal of Environmental Research and Public Health. 2020 Jan;17(11):4127.

Vivalya BM, Bin Kitoko GM, Nzanzu AK, Vagheni MM, Masuka RK, Mugizi W, Ashaba S. Affective and psychotic disorders in war-torn eastern part of the Democratic Republic of the Congo: a cross-sectional study. Psychiatry journal. 2020 Jul 24;2020.

Dossa NI, Zunzunegui MV, Hatem M, Fraser WD. Mental health disorders among women victims of conflict-related sexual violence in the Democratic Republic of Congo. Journal of Interpersonal Violence. 2015 Aug;30(13):2199-220.

Johnson K, Scott J, Rughita B, Kisielewski M, Asher J, Ong R, Lawry L. Association of sexual violence and human rights violations with physical and mental health in territories of the Eastern Democratic Republic of the Congo. Jama. 2010 Aug 4;304(5):553-62.

The study adds very little to existing knowledge base and may be of limited interest to wider readership. Perhaps a local Journal might be interested in the study.

Author response: We appreciate the reviewer’s suggestion that studies on African countries should be published in local journals. While we agree that local journals play a crucial role in promoting research and development in the region, we also believe that our findings have broad implications for the growing international research and interventions that are taking place with conflict affected populations with mental health needs. We also appreciate that the reviewer has flagged several recent studies on the broad topic of mental health in the DRC. To us this trend goes to show the broad interest in this topic over the past few years. Additionally, we would also like to flag that the studies mentioned by the reviewer were published in journals such as JAMA and Journal of Interpersonal Violence that are widely read and are not local journals, thereby indicating to us the wide readership and growing international interest in this topic.

In terms of the additional contribution our study makes, we would like to note that the studies mentioned above do not focus on the association between the two constructs of interest in our study which are ‘mental health’ and ‘work’. For example, the study by Mitonga-Monga et al. (2020) mentioned above, does not look at the same mental health constructs such as symptoms/diagnosis of PTSD, depression, anxiety and/or functioning. Furthermore, it looks at a very different population of mostly male employees with a university degree who work in the manufacturing sector. The remaining studies mentioned have only focused on traumatic events and mental health and do not explore work or productivity at all. For example, the study by Vivalya et al. (2020) only looks at the “relationship between the experience of traumatic events and onset of bipolar affective and psychotic disorders”, while the other studies mentioned above by Dossa et al. (2015) and by Johnson et al. (2010) both look at the association between mental health and sexual violence. The lack of focus on economic outcomes in the vast number of existing publications mentioned by the reviewer supports our point about how our study contributes substantially to this growing literature by continuing to look at these important mental health indicators but in the context of how they are associated with women’s work and productivity in this setting. 

Reviewer #3: The manuscript was well written and the subject is very interesting. With the context of the Democratic Republic of Congo, these data will help improve the care of women who have suffered violence. This manuscript was well written.

-The manuscript was well written and the subject is very interesting. With the context of the Democratic Republic of Congo, these data will help improve the care of women who have suffered violence.

Introduction

- The introduction is very long. We must focus on the elements that will justify the realization of this work.

-From line 113 to 121, you mention the purpose of this study, but after that comes a paragraph on the context of the study. It is better to first talk about the context before detailing the goal because it also takes into account the context of the study. The goal may be different depending on whether you work in one environment or another.

-On line 129, you cite a 2010 study without specifying the target study population.

-For a logical sequence of ideas and to respect the chronology, it would be good to move the sentence up, from line 138 to line 141, to line 136, just before starting the quote from the 2018 study.

Author response: Thank you for appreciating the manuscript and for helping us make our manuscript easier to read and help us focus on key elements. We appreciate the suggestions and have addressed them: 

- We have attempted to edit the introduction to focus on the justification of this work

- Line 113-121 have been moved to after the context.

- In what was formerly line 129 we have edited the text to specify the population “A 2010 prevalence study carried out in North and South Kivu provinces and Ituri district in the DRC found that half of the adult respondents exhibited the symptoms indicative of PTSD”

- What was originally in line 138-141 was moved before the quote from 2018 and then this section was moved up before what was line 13.

Methods

-On line 156: correct the word “nartional” by removing the “r” between “a” and “t”.

Author response: Thank you. This is corrected in the text.

-At this point, it is necessary to specify which ethics committee it is: that of the school of public health or the ministry of health.

Author response: Ethical approval was received from Health Media Lab’s Institutional Review Board with approval number IRB #: 404WBG17 on Feb 21, 2017. This is also added in the text.

-Better specify how the diagnosis of PTSD was made.

Author response: Thank you. We added the following to the manuscript: "The PTSD Symptom Scale – Interview for DSM-5 (PSS-I-5) was used to make a diagnosis of PTSD for this study (Foa, 2013)”.

- A brief description of the different sites where the study was carried out is missing.

Author response: We added the following to the manuscript: “Except for the city of Goma, the health zones in our sample are predominantly rural and poor. In each health zone, participants were identified when they sought care either in Community Based Organizations (CBOs), about 78% of the sample, or in health centers. Health centers are facilities at the lowest level of the heath system in the DRC. They also offer psychosocial assistance and legal support to survivors. NET counselors in health centers are mostly nurses and midwives. CBOs are village women's associations that offer assistance and reintegration services to survivors and socioeconomic services such as Village Savings and Loans Associations and livelihoods groups.” 

-Line 221: the "USD" must be written in full during the first quote and then abbreviated.

Author response: Thank you. This is corrected in the manuscript text.

- In the last paragraph of the methods, detail, if this has been done, the different therapeutic modalities from which the respondents would have benefited.

Author response: We added the following to the manuscript: “The “no-therapy group” received the standard-of-care provided to all women who sought care at Heal Africa, in health centers, or in CBOs. The standard-of-care includes case management services, such individual and collective counselling, referral to economic and legal support services, and health services in the case of health centers and Heal Africa.”

Results

-Line 288: modify the title: instead of saying "descriptive statistics", say rather "socio-epidemiological data".

Author response: Thank you. The title of the section has been changed to “Socio-epidemiological data” as suggested.

-Tables 7 and 8: add to the legend the meaning of Y.

Author response: Thank you for this suggestion. In the table notes “Y” stands for yes for controlling for individual fixed effects and survey round dummy. This is added in the table notes in tables 7-10.

Discussion

-Before talking about the limitations of the study, insert a paragraph to propose your model for monitoring women who have suffered violence.

Author response: Thank you. We inserted the following sentences at the end of the conclusion, which we felt was a more appropriate location for programming options to mitigate the potentially adverse mental health effects of economic programming on women: “This could take the form of a robust monitoring system of women beneficiaries’ mental and physical wellbeing. Regular data collections and the involvement of trained case managers in such activities may help detect adverse effects of economic inclusion programs. In addition, treatment for mental health disorders could be included as a component of economic programming.”

---

## [Editor Report · Decision Letter 1]

23 Mar 2023

Losing ground in the field: An exploratory analysis of the relationship between work and mental health amongst women in conflict affected Democratic Republic of the Congo

PONE-D-22-22896R1

Dear Dr. Kalra,

We’re pleased to inform you that your manuscript has been judged scientifically suitable for publication and will be formally accepted for publication once it meets all outstanding technical requirements.

Kind regards,

Tae-Young Pak, Ph.D.

Academic Editor

PLOS ONE
---

## [Editor Report · Acceptance letter]

12 Apr 2023

PONE-D-22-22896R1 

Losing ground in the field: An exploratory analysis of the relationship between work and mental health amongst women in conflict affected Democratic Republic of the Congo 

Dear Dr. Kalra:

I'm pleased to inform you that your manuscript has been deemed suitable for publication in PLOS ONE. Congratulations! Your manuscript is now with our production department. 

Kind regards, 

on behalf of

Tae-Young Pak 

Academic Editor

PLOS ONE